# The aspartyl protease DDI2 activates Nrf1 to compensate for proteasome dysfunction

Shun Koizumi[1], Taro Irie[1], Shoshiro Hirayama[1], Yasuyuki Sakurai[1], Hideki Yashiroda[1], Isao Naguro[2], Hidenori Ichijo[2], Jun Hamazaki[1], Shigeo Murata[1]*

[1]Laboratory of Protein Metabolism, Graduate School of Pharmaceutical Sciences, The University of Tokyo, Tokyo, Japan; [2]Laboratory of Cell Signaling, Graduate School of Pharmaceutical Sciences, The University of Tokyo, Tokyo, Japan

**Abstract** In response to proteasome dysfunction, mammalian cells upregulate proteasome gene expression by activating Nrf1. Nrf1 is an endoplasmic reticulum-resident transcription factor that is continually retrotranslocated and degraded by the proteasome. Upon proteasome inhibition, Nrf1 escapes degradation and is cleaved to become active. However, the processing enzyme for Nrf1 remains obscure. Here we show that the aspartyl protease DNA-damage inducible 1 homolog 2 (DDI2) is required to cleave and activate Nrf1. Deletion of DDI2 reduced the cleaved form of Nrf1 and increased the full-length cytosolic form of Nrf1, resulting in poor upregulation of proteasomes in response to proteasome inhibition. These defects were restored by adding back wild-type DDI2 but not protease-defective DDI2. Our results provide a clue for blocking compensatory proteasome synthesis to improve cancer therapies targeting proteasomes.

*For correspondence: smurata@ mol.f.u-tokyo.ac.jp

Competing interests: The authors declare that no competing interests exist.

## Introduction

Proteasome inhibition elicits a response to restore proteasome activity, or a 'bounce-back response,' where Nrf1 is the responsible transcription factor that upregulates expression of all proteasome sub-unit genes in a concerted manner in human cells (*Radhakrishnan et al., 2010*; *Steffen et al., 2010*). Proteasome inhibitors such as bortezomib and carfilzomib have been in clinical use for treatment of cancers, especially multiple myeloma, but this bounce-back response attenuates the ability of proteasome inhibitors to kill cancer cells (*Radhakrishnan et al., 2010*). Therefore, genes regulating Nrf1 activation could be useful drug targets for increasing efficacy of proteasome inhibition in cancer treatment.

When Nrf1 is produced, the bulk of the polypeptide is inserted into the ER lumen and glycosylated, with a short cytosolic N-terminus followed by a single transmembrane domain (*Radhakrishnan et al., 2014*; *Zhang et al., 2007*). The luminal region of Nrf1 is continually retrotranslocated to the cytosol by the p97/VCP ATPase complex, accompanied by deglycosylation and ubiquitination. Under normal circumstances, Nrf1 is promptly degraded by the proteasome. In contrast, when proteasome activity is compromised, Nrf1 escapes degradation and is proteolytically cleaved to the active form which enters the nucleus and enhances expression of target genes including proteasome subunits (*Radhakrishnan et al., 2014*, *2010*; *Sha and Goldberg, 2014*; *Steffen et al., 2010*). However, the processing enzyme for Nrf1 remains obscure.

**eLife digest** The proteasome is a machine that destroys unnecessary or damaged proteins inside cells. This role of the proteasome is essential for cell survival, and so when the proteasome is inhibited, cells produce new proteasomes to compensate. Upon proteasome inhibition, a protein called Nrf1 is activated and executes this "bounce-back" response. Some cancer treatments aim to kill cancer cells by inhibiting proteasomes, but these treatments may be unsuccessful if the bounce-back response is not also prevented. Therefore, understanding how Nrf1 is activated is an important issue.

Nrf1 is produced at a structure called the endoplasmic reticulum in cells and is continually destroyed by the proteasome. On the other hand, when proteasomes are inhibited, Nrf1 accumulates and is cleaved into an active form, which moves to the cell nucleus to start producing proteasomes. However, it was not known which molecule cleaves Nrf1.

Koizumi et al. set out to discover this molecule by screening the genetic material of human cells, and identified a gene that encodes a protease (an enzyme that cleaves other proteins) called DDI2. The loss of DDI2 from cells prevented Nrf1 from being cleaved and entering the nucleus, resulting in low levels of proteasome production. Further experiments showed that a mutant form of DDI2 that lacked protease activity was unable to cleave Nrf1, confirming DDI2's role in activating Nrf1.

Deleting DDI2 from cells does not completely prevent the cleavage of Nrf1, and so some other cleaving enzyme might exist; the identity of this enzyme remains to be discovered. Future work is also needed to establish exactly how DDI2 cleaves Nrf1. This could help to develop a DDI2 inhibitor for cancer treatment that could be used in combination with existing proteasome inhibitors.

## Results and discussion

To identify genes important for Nrf1 activation, we performed a genome-wide small interfering RNA (siRNA) screen (*Figure 1—figure supplement 1A*). Our approach used the well-characterized sub-cellular localization of Nrf1 accumulation in the nucleus in response to proteasome inhibition. HEK293A cells were transfected with pooled siRNA (a pool of 4 unique siRNAs per gene) and then treated with the proteasome inhibitor bortezomib to induce accumulation and nuclear translocation of Nrf1 (*Steffen et al., 2010*). Cells were then fixed and stained with anti-Nrf1 antibody. The ratio of the nuclear to cytoplasmic fluorescent intensities was assessed by high-content microscopy and automated image analysis (*Figure 1—figure supplement 1B*). p97 siRNA treatment served as a positive control, which abolished Nrf1 translocation following bortezomib treatment while increasing cytoplasmic Nrf1 (*Figure 1A*) (*Radhakrishnan et al., 2014*). We observed a high degree of assay robustness (Z'-factor > 0.5, *Figure 1—figure supplement 1C*) in the primary screen. The initial candidate genes with B score < –3.2 (*Figure 1B*) or which were picked up by visual inspection of the raw image data were further tested using four individual siRNAs in two different cell lines (HEK293A and HT1080 cells) (*Figure 1C*). The subsequent candidates that had more than two hits in either cell line were finally examined whether the siRNAs mitigated upregulation of *PSMA3*, a proteasome sub-unit gene. Consequently, we obtained 14 candidate genes that may impair activation of Nrf1 in response to bortezomib treatment (*Figure 1D*). These hits included SEL1L, a co-factor of the ubiquitin ligase HRD1, which catalyzes ER-associated degradation (ERAD) of Nrf1 (*Iida et al., 2011*; *Steffen et al., 2010*; *Tsuchiya et al., 2011*) and FAF2/UBXD8, a p97-recruiting molecule in ERAD (*Meyer et al., 2012*), validating our screening approach (*Figure 1A,B, and D*).

Among the final hit genes, we focused on DDI2, because it has a typical retroviral aspartyl protease domain, and therefore is a candidate Nrf1 processing enzyme (*Krylov and Koonin, 2001*). In negative and positive (p97 siRNA) control cells treated with bortezomib, the majority of Nrf1 is localized in the nucleus and the cytoplasm, respectively (*Figure 2A*). DDI2 knockdown partially inhibited nuclear translocation of Nrf1, accompanied by an increase in cytoplasmic Nrf1. To determine if there is a defect in Nrf1 processing by DDI2 knockdown, we examined which Nrf1 species were observed in each knockdown. In negative control cells, Nrf1 was hardly detected in the absence of bortezomib, but bortezomib treatment increased the processed, active form of Nrf1 as well as the full-length, cytosolic form that is retrotranslocated into the cytosol by p97 (*Figure 2B*)

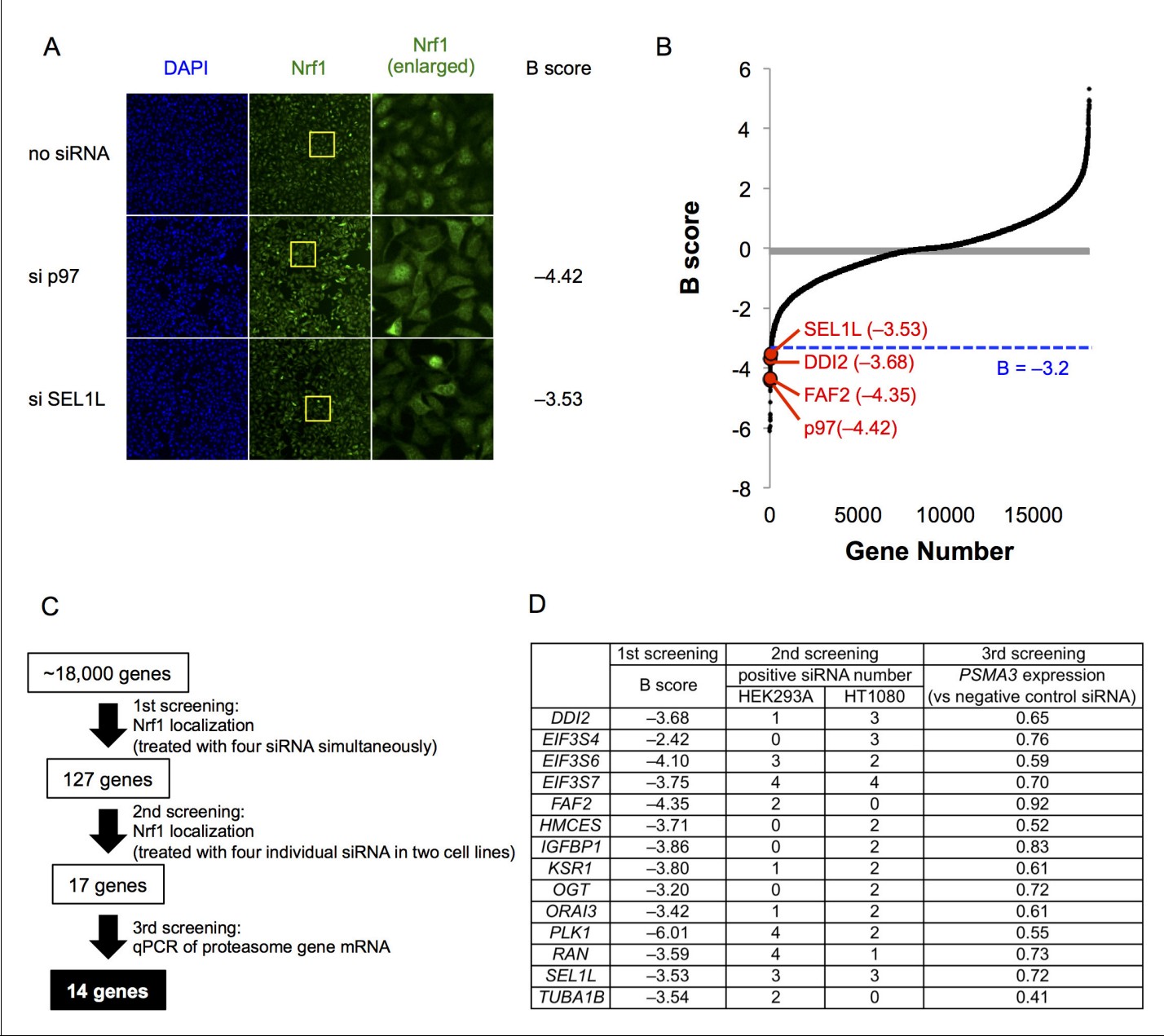

**Figure 1.** A genome-wide siRNA screen for regulators of Nrf1 translocation to the nucleus in response to proteasome inhibition. (**A**) Representative images of Nrf1 localization in control cells (no siRNA) and cells transfected with siRNA targeting p97 or SEL1L in the primary screen. Yellow-boxed regions are magnified and displayed in the right panels. (**B**) B score of all samples in the primary screen. Data are ordered from lowest to highest. Dashed blue line represents a cutoff value for positive hits. Some of the representative final hits are shown as red dots. The list of B scores for all samples in the primary screen are available in the *Figure 1—source data 1*. (**C**) Workflow and summary of the genome-wide siRNA screen. (**D**) List of the 14 final hit genes and the score in each assay throughout the screening process.

The following source data and figure supplement are available for figure 1:

**Source data 1.** List of B-score in the primary screen.

**Figure supplement 1.** Methods for the genome-wide screen.

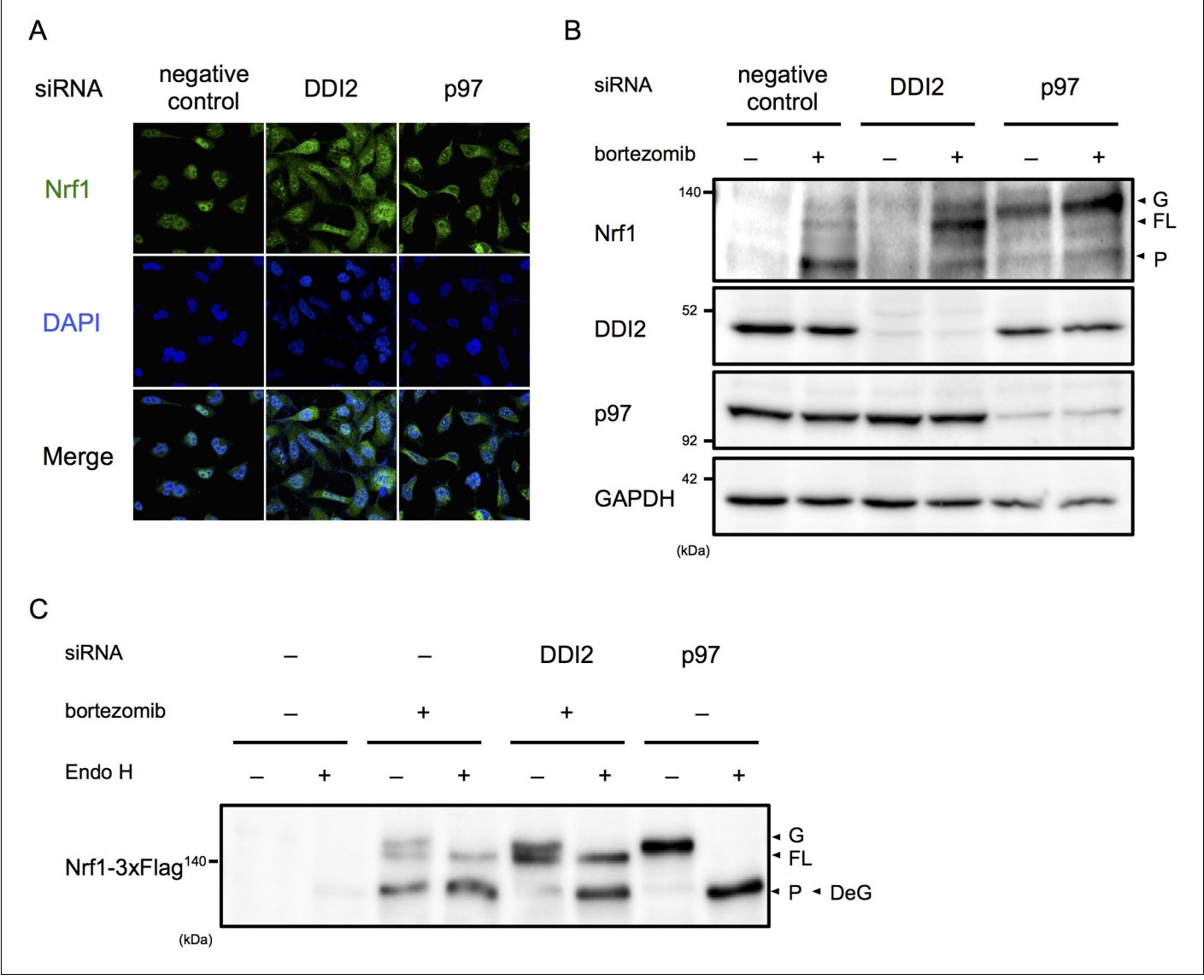

**Figure 2.** DDI2 is involved in Nrf1 processing and translocation to the nucleus. (**A**) Representative images of Nrf1 localization. HEK293A cells were transfected with a non-targeting control (negative control), DDI2, or p97 siRNA and then treated with 50 nM bortezomib for 14 hr before fixation. (**B**) Immunoblotting of whole-cell lysates of cells in (**A**) treated with or without bortezomib. Nrf1 is detected as three different forms; a glycosylated form (G), full-length form (FL), and processed form (P). (**C**) Immunoblotting of Nrf1 after deglycosylation treatment. HEK293A cells were transfected with DDI2 or p97 siRNA, followed by transfection with Nrf1-3×Flag, and then treated with or without 50 nM bortezomib. The cell lysates were treated with or without Endo H. DeG denotes deglycosylated Nrf1.

The following figure supplement is available for figure 2:

**Figure supplement 1.** The expression and localization of DDI2 were not affected by bortezomib treatment.

(*Radhakrishnan et al., 2014*). In p97 knockdown cells, the luminal ER form of Nrf1 that is N-glycosylated accumulated while the processed form almost disappeared both in the presence and absence of bortezomib. In DDI2 knockdown cells, Nrf1 was not detected in the absence of bortezomib, similar to control cells. However, in the presence of bortezomib, the full-length, cytosolic form of Nrf1 markedly accumulated (*Figure 2B*). These results indicate that DDI2 is involved in conversion of full-length, cytosolic Nrf1 to the processed, active form.

We further examined the N-glycosylation status of Nrf1 using cells transfected with Nrf1 tagged with 3×Flag at the C-terminus. N-glycosylated Nrf1 accumulated in p97 knockdown cells was sensitive to endoglycosidase H (Endo H) treatment and the deglycosylated form migrated faster in SDS-PAGE, consistent with a previous report (*Figure 2C*) (*Radhakrishnan et al., 2014*). In contrast, the full-length form of Nrf1 that is significantly accumulated in DDI2 knockdown cells was not sensitive to Endo H treatment (*Figure 2C*). Note that bortezomib treatment alone causes some accumulation of Endo H-sensitive N-glycosylated Nrf1 and that the deglycosylated species was detected at almost the same molecular weight as the processed, active form of Nrf1 observed in cells treated with bortezomib alone, similar to previously reported observations (*Figure 2C*) (*Radhakrishnan et al., 2014*). These results demonstrate that the form of Nrf1 accumulated in DDI2-depleted cells is not N-glycosylated, further supporting the role of DDI2 in the processing of Nrf1 rather than in deglycosylation or retrotranslocation.

The X-ray crystal structure analysis of the retroviral aspartyl protease (RVP) domain of budding yeast Ddi1p has revealed that it is a dimer with a similar fold to that of the human immunodeficiency virus type 1 (HIV-1) protease, with identical geometry of the double D[S/T]GA motif of the active site (*Sirkis et al., 2006*). The HIV-1 protease typically cleaves substrates between two hydrophobic residues (*Konvalinka et al., 2015*). Nrf1 has been shown to be cleaved between Trp103 and Leu104 to become active (*Radhakrishnan et al., 2014*), which conforms with the cleavage motif by retroviral aspartyl proteases. Accordingly, we asked whether the protease activity of DDI2 is required for Nrf1 processing. DDI2 has a ubiquitin-like domain (UBL) at the N-terminus and a RVP domain near the C-terminus (*Figure 3A*). Bortezomib treatment increased the processed form of Nrf1 (*Figure 3B*). Knockdown of DDI2 reduced the processed form and increased full-length Nrf1 (*Figure 3B*). This effect was rescued by introducing siRNA-resistant wild-type DDI2 but not a protease-dead DDI2 in which the active site aspartic acid 252 was replaced with asparagine (D252N). We also found that a DDI2 mutant lacking the UBL domain only partially restored the effect of DDI2 knockdown (*Figure 3B*). These results suggest that the protease activity of DDI2 is required for cleavage of Nrf1 and that the UBL domain plays some role in the cleavage.

To confirm the necessity of the protease activity of DDI2, we generated DDI2 knockout (KO) and protease-dead DDI2 (D252N) knock-in (KI) cells as well as wild-type DDI2 knock-in cells (*Figure 3*, *Figure 3—figure supplement 1A,B*). In DDI2 knockout cells and DDI2 D252N knock-in cells, the full-length form of Nrf1 was accumulated upon bortezomib treatment, whereas the processed form was accumulated in wild-type DDI2 knock-in cells (*Figure 3C*). These results further support the requirement of DDI2 protease activity for Nrf1 activation.

We then examined whether a lack of the catalytic activity of DDI2 abolishes the 'bounce-back' response after proteasome inhibition. In parental HCT116 cells, bortezomib treatment caused an increase in mRNA levels of the proteasome subunit genes *PSMA3* and *PSMB5* (*Figure 3D*). Knockout of DDI2 strongly suppressed this response, further supporting the importance of DDI2 in Nrf1 activation (*Figure 3D*). Interestingly, the basal expression of proteasome subunits was also decreased in DDI2-deficient cells. In wild-type DDI2 knock-in cells, mRNA levels of the proteasome subunits were upregulated in response to bortezomib, similar to the parental cells. In contrast, DDI2 D252N knock-in cells did not undergo such a response, similar to DDI2 knockout cells (*Figure 3D*). These results suggest that the processing of Nrf1 by the aspartyl protease activity of DDI2 is required for upregulation of proteasome gene expression mediated by Nrf1 in response to proteasome inhibition.

Nrf1 has also been found to regulate basal expression of proteasome subunits, the extent of which varies between cell types (*Lee et al., 2013*, *2011*). We observed that knockout and D252N DDI2 knock-in cells had significantly lower proteasome activity compared to wild-type DDI2 knock-in cells, suggesting that DDI2 is also involved in basal expression of proteasomes through its catalytic activity (*Figure 3E*).

In conclusion, we identified DDI2 as a protease that is required for Nrf1 processing and the bounce-back response induced by proteasome inhibition. However, there remain several questions to be answered. How can the involvement of DDI2 be reconciled with a previous report that demonstrated a defect in Nrf1 processing by strong inhibition of the proteasome, leading to the conclusion that the proteasome is the processing enzyme for Nrf1 (*Sha and Goldberg, 2014*)? In terms of substrate specificity, the cleavage site of Nrf1 (P1: W, P1′: L) does not seem to be a sequence preferred by the proteasome (*Toes et al., 2001*); rather it conforms to a cleavage motif of RVP (*Konvalinka et al., 2015*). It could be that the proteasome activity is required for function of DDI2 or

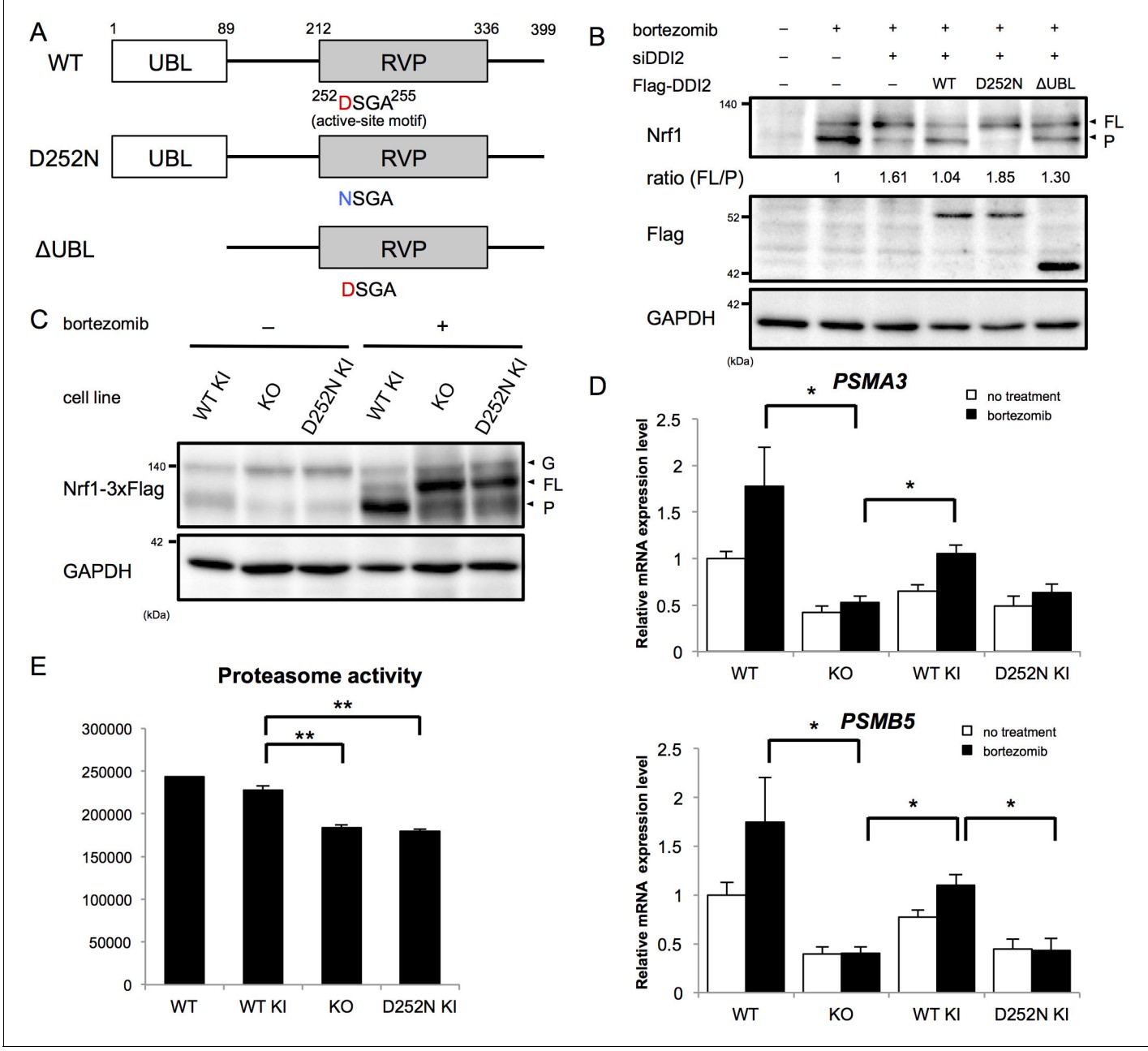

**Figure 3.** The protease activity of DDI2 is necessary for Nrf1 processing and its transcriptional activity. (A) Schematic diagram of wild-type (WT) and each mutant of DDI2. Ubiquitin-like (UBL) domain and retroviral protease-like (RVP) domain are represented as filled rectangles. The putative aspartyl protease active site amino acid sequence is shown. (B) HEK293A cells were transfected with DDI2 siRNA and after 24 hr were transfected with a plasmid encoding WT or mutant DDI2 shown in (A), followed by 50 nM bortezomib treatment for 14 hr before harvest. The signal intensity ratio of Nrf1 full-length form (FL) to the processed form (P) was calculated, where the ratio for bortezomib treatment alone was set as 1. (C) Immunoblotting of whole cell lysates of DDI2 WT knock-in (KI), DDI2 knockout (KO), and DDI2 D252N KI HCT116 cells. The cells transfected with Nrf1-3×Flag were treated with or without 50 nM borteaomib. (D) Relative mRNA expression of the proteasome genes *PSMA3* and *PSMB5* in WT, DDI2 KO, DDI2 WT KI, and DDI2 D252N KI HCT116 cells. mRNA levels of target genes were normalized by *GUSB* mRNA levels. The data represent mean + standard error of the mean (SEM) (n = 3, biological replicates). Statistical comparison was made by Tukey's test (*p<0.05). (E) Proteasome peptidase activity of cell lysates of the indicated cell lines. The data represent mean + SEM (n = 3, biological replicates). Statistical comparison was made by Tukey's test (**p<0.01).

The following figure supplement is available for figure 3:

**Figure supplement 1.** Genome editing of *DDI2* locus by CRISPR-Cas9 system.

some other factors that is involved in Nrf1 processing. Related to this, the mechanism by which DDI2 acts as a Nrf1 processing protease remains unclear. DDI2 is not induced by bortezomib at either protein or mRNA level (*Figure 2B* and *Figure 2—figure supplement 1A*). Furthermore, the subcellular localization of DDI2 seems to be unaffected by bortezomib treatment (*Figure 2—figure supplement 1B*). Since DDI2 is suggested to be active even when the proteasome activity is not compromised (*Figure 3D and E*), a specific activation mechanism under proteasome impairment may not exist. An intriguing finding is that the UBL domain of DDI2 plays some role in Nrf1 processing (*Figure 3B*). It has been shown that the UBL domain of Ddi1p is an atypical UBL that binds ubiquitin (*Nowicka et al., 2015*). Binding of DDI2 with ubiquitinated proteins, possibly Nrf1 itself, would be promoted by proteasome inhibition and may facilitate Nrf1 processing by DDI2. Lastly, whether DDI2 directly cleaves Nrf1 remains unknown. We have tested a recombinant fragment of Nrf1 encompassing the processing site as a substrate for recombinant DDI2, but failed to detect its cleavage. Other factors might be required for in vitro reconstitution of Nrf1 processing by DDI2, such as substrate unfolding, co-activators of DDI2, and a set of specific experimental conditions. Understanding the mechanism by which DDI2 cleaves Nrf1 and establishing an in vitro assay for the enzymatic activity of DDI2 should provide useful information for developing a DDI2 inhibitor that would block compensatory proteasome synthesis to improve cancer therapies targeting proteasomes.

## Materials and methods

### Genome-wide siRNA screening

In the primary screen, Dharmacon siGENOME SMARTpool siRNA library (GE Dharmacon, Lafayette, CO) was used. To prepare screening plates, the siRNAs in each well were suspended in $1 \times$ siRNA buffer (Thermo Fisher Scientific, Waltham, MA) and 2.5 pmol siRNA (2.5 µL/well) was dispensed into black, clear bottom, 384-well plates (Greiner, Kremsmünster, Austria). For each well, a mixture of 10 µL DMEM and 0.1 µL Lipofectamine RNAiMAX (Invitrogen, Carlsbad, CA) was added. After 40 min incubation, 2000 cells/well of HEK293A cells were seeded. After 48 hr culture, bortezomib was added into each well to a final concentration of 10 nM. Cells were fixed with 4% PFA after 12 hr bortezomib treatment. Cells were then stained with Nrf1 antibody (sc-13031; Santa Cruz Biotechnology, Dallas, TX) and DAPI, and the fluorescent images were acquired and analyzed by CellInsight High Content Screening Platform (Thermo Fisher Scientific). The fluorescence signal ratio of the nucleus to the cytoplasm was used as a raw measured value. The value was fitted in a two-way median polish method to exclude positional effects in the 384-well plate, and then the B score was calculated on a per-plate basis using the following formula.

$$\text{B score} = (X_i - \text{Median}) / \text{MAD}$$

$$(X_i: \text{ measured value, MAD} : \text{ median of absolute deviation})$$

In the secondary screen, four individual siRNAs contained in the library were purchased from Dharmacon and used. HEK293A and HT1080 cells transfected with each siRNA were analyzed by the same method as in the primary screening. In the third screen, HEK293A cells were treated with each hit siRNA and the expression level of the proteasome gene *PSMA3* was measured by quantitative RT-PCR.

### DNA constructs

Human DDI2 cDNA was synthesized from total RNA extracted from HT1080 cells using the indicated primers. Forward: 5'-ATGCTGCTCACCGTGTACTGTGTGC-3', Reverse: 5'-TCATGGCTTCTGACGC TCTGCATCC-3'. DDI2 UBL deletion mutant was synthesized using 5'-AACTTACCCCGAATAGA TTTCAG-3' for a forward primer. siRNA resistant mutations were introduced without changing amino acid sequence using the following primers. Forward: 5'- TAATGTTGTATATTAACTGCAAAGTGAA TGGACATCCTG-3', Reverse: 5'- CGACCTGTCCAAAACTTTCCGGAGCCTCTTCCATAGC-3'. Human Nrf1 cDNA was synthesized from total RNA extracted from HEK293A cells using the indicated primers. Forward: 5'-ATGCTTTCTCTGAAGAAATACTTAACG-3', Reverse: 5'-TCACTTTCTCCGGTCC

TTTGG-3'. PCR was performed using PrimeSTAR Max DNA polymerase (Takara Bio, Shiga, Japan). Amplified fragments were subcloned into pIRES vector (Clontech Laboratories, Mountain View, CA) and all constructs were confirmed by sequencing.

## Cell culture and transfection

HEK293A cells were purchased from Thermo Fisher Scientific. HCT116 cells were obtained from RIKEN BRC. The cell lines were tested negative for mycoplasma contamination by DAPI staining. The authors performed no further authentication of the cell lines. HEK293A cells and HCT116 cells were cultured under standard conditions. cDNAs were transfected into cells using PEI-MAX (Mw: 40,000). siRNAs targeting DDI2 or p97 and siGENOME Non-Targeting siRNA #2 were purchased from GE Dharmacon. siRNA was transfected into cells with Lipofectamine RNAi MAX (Invitrogen). The sequences of siRNAs targeting DDI2 and p97 were as follows: DDI2, 5'-GCCAAGUAGUGAUGC UUUA-3'; p97, 5'-GUAAUCUCUUCGAGGUAUA-3'.

## Establishment of DDI2 knockout and knock-in cell lines

The cell lines were established using the CRISPR/Cas9 system. Single guide RNAs (sgRNA) were designed using CRISPR direct (http://crispr.dbcls.jp/) and cloned into a pX330 vector. The sgRNA sequence for *DDI2* was 5'-ACTCGAGCTCGCACAGCGCG-3'. Targeting constructs for gene knock-out were designed to insert a puromycin resistance cassette at the locus of the start codon. Targeting constructs for DDI2 knock-in were designed to insert DDI2 wild type or D252N cDNA in-frame downstream of DDI2 exon 1. A puromycin resistance cassette was also inserted into this region. The sgRNA vector and targeting vector were transfected in HCT116 cells. After 48 hr transfection, cells were cultured in medium supplemented with 4 µg/mL puromycin. After two weeks drug selection, colonies were picked up and successful homologous recombination was confirmed by PCR method. PCR was performed using EmeraldAmp PCR Master Mix (Takara Bio). The following primers were used for confirmation of genome editing. DDI2 Forward: 5'-ATGCTGCTCACCGTGTACTGTGTGC-3', DDI2 intron 1 Reverse: 5'-GCAAGCTGAGTAGGGAAATGAAACCACCAA-3', Puro forward: 5'-G TCACCGAGCTGCAAGAACTCTTCC-3'.

## Quantitative RT-PCR

Cells were harvested 12 hr after 20 nM bortezomib treatment. Total RNA of cells was isolated using High Pure RNA isolation kit (Roche, Basel, Switzerland) and were reverse-transcribed using ReverTra Ace qPCR RT kit (Toyobo, Osaka, Japan). Quantitative RT-PCR was performed using THUNDERBIRD Probe qPCR Mix (Toyobo), Universal ProbeLibrary Probe (Roche), and LightCycler 480 (Roche). The sequences of primers used were as follows: *PSMA3*, 5'-GAAGAAGCAGAGAAATATGCTAAGG-3' and 5'-GGCTAAATAGTTACATTGGACTGGAG-3'; *PSMB5*, 5'-CATGGGCACCATGATCTGT-3' and 5'-GAAATCCGGTTCCCTTCACT-3'; *GUSB*, 5'-CGCCCTGCCTATCTGTATTC-3' and 5'-TCCCCA-CAGGGAGTGTGTAG-3'.

## Immunoblotting

24 hr after transfection of siRNA, cells were transfected with cDNA and cultured for a further 48 hr. 50 nM bortezomib was added 14 hr prior to cell lysis. Cells were lysed in buffer containing 42 mM Tris-HCl (pH 6.8), 1.72% SDS, 5.6% glycerol, 5% 2-mercaptoethanol, and 0.01% bromophenol blue (SDS sample buffer) for whole-cell lysate. The samples were subjected to SDS-PAGE, transferred to polyvinylidene fluoride membrane, and analyzed by immunoblotting. All images were taken using Fusion SL4 (M&S Instruments). Rabbit polyclonal antibody against DDI2 was raised by immunizing keyhole limpet hemocyanin (KLH) conjugated synthetic DDI2 C-terminal (residues 385–399) peptides. The following antibodies were purchased: Nrf1 (sc-13031; Santa Cruz), p97 (MA3-004; Invitrogen), GAPDH (sc-32233; Santa Cruz), Flag (F1804; Sigma Aldrich, St. Louis, MO).

## Immunostaining

Cells were fixed in 4% paraformaldehyde 72 hr after transfection of siRNA and 16 hr after 10 nM bortezomib treatments. The cells were incubated with primary antibodies, and then incubated with DAPI (Nacalai Tesque, Kyoto, Japan) and secondary antibodies, either Goat anti-rabbit or anti-

mouse IgG secondary antibody Alexa Fluor 488 or Alexa Fluor 647 conjugate (Invitrogen). All images were acquired by TCS SP5 or TCS SP8 (Leica Microsystems, Wetzlar, Germany).

## Proteasome activity measurement

Cells were lysed in ice-cold buffer containing 25 mM Tris-HCl (pH 7.5), 0.2% Nonidet P-40, 1 mM dithiothreitol, 2 mM ATP, and 5 mM $MgCl_2$. The hydrolysis of the fluorogenic peptide, succinyl-Leu-Leu-Val-Tyr-7-amino-4-methylcoumarin (Suc-LLVY-MCA) (Peptide Institute, Osaka, Japan) was measured in 50 mM Tris-HCl (pH 8.0) at 37°C by ARVO MX 1420 (PerkinElmer, Waltham, MA).

## Deglycosylation assay

Cells were lysed in ice-cold phosphate buffered saline (PBS) containing 0.5% Triton X-100. After centrifugation (20,000 g, 10 min), the cell lysates were subjected to deglycosylation reactions with Endo $H_f$ (New England BioLabs, Ipswich, MA) following the manufacturer's protocol.

## Statistical analysis

A biological replicate was considered as each independent experiment. Each different clone of the same genotypes was also considered as a biological replicate in the experiments using mutant cell lines obtained by the CRISPR-Cas9 system. Technical replicates were multiple analyses of the same sample in an experiment. The results are expressed as mean + standard error of the mean (SEM) of three biological replicates (n = 3). Significant differences were considered as probabilities less than 5% ($p < 0.05$).

## Acknowledgements

We are grateful to the members of the Murata laboratory for helpful advice and discussion and the members of the Ichijo laboratory for experimental support. This work was funded by JSPS grants 25221102 and 26000014 to SM.

## Additional information

### Funding

| Funder | Grant reference number | Author |
|---|---|---|
| Japan Society for the Promotion of Science | 25221102 | Shigeo Murata |
| Japan Society for the Promotion of Science | 26000014 | Shigeo Murata |
| Takeda Science Foundation | | Shigeo Murata |

The funders had no role in study design, data collection and interpretation, or the decision to submit the work for publication.

### Author contributions

SK, TI, JH, Conception and design, Acquisition of data, Analysis and interpretation of data; SH, YS, HY, IN, HI, Acquisition of data, Analysis and interpretation of data; SM, Conception and design, Analysis and interpretation of data, Drafting or revising the article

### Author ORCIDs

Shigeo Murata, http://orcid.org/0000-0002-3177-3503

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
