## [Decision Letter]

Thank you for submitting your article "The aspartyl protease DDI2 activates Nrf1 to compensate for proteasome dysfunction" for consideration by *eLife*. Your article has been reviewed by three peer reviewers, and the evaluation has been overseen by Ivan Dikic as the Senior and Reviewing Editor. The following individuals involved in review of your submission have agreed to reveal their identity: Raymond J Deshaies (Reviewer #1) and Anne Bertolotti (Reviewer #2).

The reviewers have discussed the reviews with one another and the Reviewing Editor has drafted this decision to help you prepare a revised submission.

Summary:

The manuscript by Koizumi et al. identifies DDI2 as a candidate enzyme that mediates the cleavage and activation of the transcription factor Nrf1 (NFE2L1). The manuscript starts with an unbiased genome-wide siRNA screen aimed at identifying genes important for Nrf1 activation. This yielded 14 candidate genes, including DDI2, which is known to have an aspartyl protease domain homologous to that of the HIV protease. Subsequent experiments demonstrated diminished nuclear localization and processing of Nrf1, and diminished induction of proteasome genes, in DDI2-deficient cells treated with proteasome inhibitor. It was also shown that the active site of DDI2 is required to promote processing of Nrf1 and bortezomib-induced expression of proteasome genes. The authors also showed that DDI2 can be targeted by inhibitors of HIV protease and that these inhibitors improve the efficacy of Bortezomib.

The findings presented in this manuscript are technically convincing and targeting the bounce-back response upon proteasome inhibition is indeed of importance at several levels: to improve our knowledge on the regulation of proteasome upregulation and to improve cancer therapies based on proteasome inhibition.

Essential revisions:

1) Did the authors attempt a protease protection experiment to examine the disposition of Nrf1 in DDI2-depleted cells? Based on the data Figure 2, the authors conclude that Nrf1 has been retrotranslocated and deglycosylated in DDI2-depleted cells, but this is not formally proven. It is possible that DDI2 regulates Nrf1 in some other manner that is upstream of the processing step (e.g. retrotranslocation), and the MW shift occurs for some other reason other than deglycosylation. While this may be unlikely, it is certainly possible and should be commented.

2) Figure 4 is weak, and all reviewers felt that needs to be significantly improved or removed. This is not a point that is strictly necessary for the central finding of the paper that DDI2 is required for processing. The real question is whether it can be improved by additional data (see below) or should be removed from the current manuscript. For example, the effects in panel A are weak and would benefit from a titration to higher concentrations. Panel B is even weaker. Other groups have reported that there is some synthetic effect of combining proteasome inhibitors with HIV protease inhibitors (e.g. the Kraus and Driessen papers cited in the Results and Discussion). So, what is new about this figure? The big question is, what is the underlying mechanism? Is it really due to inhibition of DDI2? A couple of questions related to this experiment immediately come to mind. Are DDI2∆ cells, like NRF1∆ cells, more sensitive to bortezomib? Is the synthetic effect of combining bortezomib and NFV or LPV still observed in DDI2∆ cells? Is it observed in NRF1∆ cells? So we recommend to the authors to decide one of these two options related to Figure 4.

3) The authors have established DDI2 knock-out cells (DDI2 KO) and DDI2 KO cells in which the WT or protease-dead mutant DDI2 cDNA was knock-in to the DDI2 locus. It would be useful to analyse the processing of Nrf1 in these cell lines upon proteasome inhibition to complement the siRNA knocked down experiments.

4) The authors should discuss why both forms (FL and P) of Nrf1 are decreasing after co-treatment with bortezomib and lopinavir. Could it be due to cell death or off-target effect of this drug? Is lopinavir treatment alone also decreasing the levels of Nrf1?

---

## [Author Response]

*Essential revisions:*

*1) Did the authors attempt a protease protection experiment to examine the disposition of Nrf1 in DDI2-depleted cells? Based on the data Figure 2, the authors conclude that Nrf1 has been retrotranslocated and deglycosylated in DDI2-depleted cells, but this is not formally proven. It is possible that DDI2 regulates Nrf1 in some other manner that is upstream of the processing step (e.g. retrotranslocation), and the MW shift occurs for some other reason other than deglycosylation. While this may be unlikely, it is certainly possible and should be commented.*

We examined whether the Nrf1 species accumulated in DDI2-depleted cells is deglycosylated using endoglycosidase H (Endo H), which removes N-linked glycosylation. In p97-depleted cells, the accumulated Nrf1 band was sensitive to Endo H treatment, consistent with the previous studies and indicating that the two bands correspond to glycosylated “G” and deglycosylated “DeG” forms of Nrf1. In contrast, the Nrf1 form “FL” accumulated in DDI2-depleted cells is not sensitive to Endo H treatment, indicating that this form is not glycosylated and that the “FL” form is a product after retrotranslocation. Curiously, the migration of “DeG” and “FL” does not correspond to each other. This phenomenon was also observed in a previous report, though the precise reason is yet unknown (Radhakrishnan et al. 2014). These results are newly added as Figure 2 and described in the third paragraph of the Results and Discussion in the revised manuscript.

We also performed protease protection experiments to examine whether Nrf1 retrotranslocation was inhibited, but we were unable to obtain clear results. However, DDI2-depleted cells show accumulation of deglycosylated Nrf1, in contrast to p97-depleted cells (Figure 2). As deglycosylation of ER-associated degradation substrates are generally completed in the cytoplasm after retrotranslocation from the ER, this result suggests that Nrf1 is retrotranslocated in DDI2-depleted cells.

*2) Figure 4 is weak, and all reviewers felt that needs to be significantly improved or removed. This is not a point that is strictly necessary for the central finding of the paper that DDI2 is required for processing. The real question is whether it can be improved by additional data (see below) or should be removed from the current manuscript. For example, the effects in panel A are weak and would benefit from a titration to higher concentrations. Panel B is even weaker. Other groups have reported that there is some synthetic effect of combining proteasome inhibitors with HIV protease inhibitors (e.g. the Kraus and Driessen papers cited in the Results and Discussion). So, what is new about this figure? The big question is, what is the underlying mechanism? Is it really due to inhibition of DDI2? A couple of questions related to this experiment immediately come to mind. Are DDI2∆ cells, like NRF1∆ cells, more sensitive to bortezomib? Is the synthetic effect of combining bortezomib and NFV or LPV still observed in DDI2∆ cells? Is it observed in NRF1∆ cells? So we recommend to the authors to decide one of these two options related to Figure 4.*

We decided to remove Figure 4 from this manuscript after performing some experiments related to this figure. We examined bortezomib sensitivity of DDI2 knock-in and knock-out cells as well as Nrf1 knock-out cells. Although DDI2 knock-out cells tended to be more sensitive to bortezomib than DDI2 wild-type knock-in cells, the effect was not significant. Also, at least in our hands, Nrf1 knock-out, which has been shown to increase bortezomib sensitivity, did not confer significant sensitivity to bortezomib in HCT116 cells. As it is known that contribution of Nrf1 on proteasome expression depends on cell types, the results might not be generalized. We also examined whether a synergistic effect of bortezomib and LPV or NFV was observed in DDI2 knock-out or Nrf1 knock-out cells. The synergistic effect was slightly decreased but still observed in both cell lines. Thus, the HIV protease inhibitor could inhibit Nrf1 processing via DDI2 as suggested in our previous Figure 4B. However, since the decrease in the synergistic effect was so modest that we cannot conclude that synergistic cytotoxicity in combination treatment with bortezomib is mainly caused by Nrf1 or DDI2. Accordingly, we deleted sentences regarding Figure 4 in the revised manuscript.

*3) The authors have established DDI2 knock-out cells (DDI2 KO) and DDI2 KO cells in which the WT or protease-dead mutant DDI2 cDNA was knock-in to the DDI2 locus. It would be useful to analyse the processing of Nrf1 in these cell lines upon proteasome inhibition to complement the siRNA knocked down experiments.*

We examined Nrf1 processing in DDI2 knockout, DDI2 WT knock-in, and DDI2 D252N knock-in cells, and the result is newly inserted in Figure 3. Defect of Nrf1 cleavage is observed in DDI2 knockout cells and DDI2 D252N knock-in cells as in Figure 3, in which the cells were treated with DDI2 siRNA and transfected with D252N cDNA. This results support the idea that DDI2 is a processing enzyme of Nrf1.

*4) The authors should discuss why both forms (FL and P) of Nrf1 are decreasing after co-treatment with bortezomib and lopinavir. Could it be due to cell death or off-target effect of this drug? Is lopinavir treatment alone also decreasing the levels of Nrf1?*

The decrease of Nrf1 after co-treatment with bortezomib and lopinavir is reproducible, but the reason is not clear. Cell death contributed to this result to some extent because co-treatment with bortezomib and 10 μM lopinavir caused cell death. However, Nrf1 was also decreased in combination treatment with bortezomib and 1 μM lopinavir, though cell death was not induced in this condition. This effect was not observed in co-treatment with bortezomib and nelfinavir, suggesting that this was an off-target effect of lopinavir. It is possible that lopiavir itself could decrease the amount of Nrf1. However, it was not confirmed because Nrf1 was hardly detected without proteasome inhibitor. As we decided to delete Figure 4, we did not include discussion regarding this point.